# Transportability from Multiple Environments with Limited Experiments: Completeness Results

**Elias Bareinboim**
Computer Science
UCLA
eb@cs.ucla.edu

**Judea Pearl**
Computer Science
UCLA
judea@cs.ucla.edu

## Abstract

This paper addresses the problem of $mz$-transportability, that is, transferring causal knowledge collected in several heterogeneous domains to a target domain in which only passive observations and limited experimental data can be collected. The paper first establishes a necessary and sufficient condition for deciding the feasibility of $mz$-transportability, i.e., whether causal effects in the target domain are estimable from the information available. It further proves that a previously established algorithm for computing transport formula is in fact complete, that is, failure of the algorithm implies non-existence of a transport formula. Finally, the paper shows that the do-calculus is complete for the $mz$-transportability class.

## 1 Motivation

The issue of generalizing causal knowledge is central in scientific inferences since experiments are conducted, and conclusions that are obtained in a laboratory setting (i.e., specific population, domain, study) are transported and applied elsewhere, in an environment that differs in many aspects from that of the laboratory. If the target environment is arbitrary, or drastically different from the study environment, no causal relations can be learned and scientific progress will come to a stand-still. However, the fact that scientific experimentation continues to provide useful information about our world suggests that certain environments share common characteristics and that, owed to these commonalities, causal claims would be valid even where experiments have never been performed.

Remarkably, the conditions under which this type of extrapolation can be legitimized have not been formally articulated until very recently. Although the problem has been extensively discussed in statistics, economics, and the health sciences, under rubrics such as "external validity" [1, 2], "meta-analysis" [3], "quasi-experiments" [4], "heterogeneity" [5], these discussions are limited to verbal narratives in the form of heuristic guidelines for experimental researchers – no formal treatment of the problem has been attempted to answer the practical challenge of generalizing causal knowledge across multiple heterogeneous domains with disparate experimental data as posed in this paper. The lack of sound mathematical machinery in such settings precludes one of the main goals of machine learning (and by and large computer science), which is automating the process of discovery.

The class of problems of causal generalizability is called *transportability* and was first formally articulated in [6]. We consider the most general instance of transportability known to date that is the problem of transporting experimental knowledge from heterogeneous settings to a certain specific target. [6] introduced a formal language for encoding differences and commonalities between domains accompanied with necessary or sufficient conditions under which transportability of empirical findings is feasible between two domains, a source and a target; then, these conditions were extended for a complete characterization for transportability in one domain with unrestricted experimental data [7, 8]. Subsequently, assumptions were relaxed to consider settings when only limited experiments are available in the source domain [9, 10], further for when multiple source domains

with unrestricted experimental information are available [11, 12], and then for multiple heterogeneous sources with limited and distinct experiments [13], which was called "$mz$-transportability".[1]

Specifically, the $mz$-transportability problem concerns with the transfer of causal knowledge from a heterogeneous collection of source domains $\Pi = \{\pi_1, ..., \pi_n\}$ to a target domain $\pi^*$. In each domain $\pi_i \in \Pi$, experiments over a set of variables $\mathbf{Z}_i$ can be performed, and causal knowledge gathered. In $\pi^*$, potentially different from $\pi_i$, only passive observations can be collected (this constraint will be weakened). The problem is to infer a causal relationship $R$ in $\pi^*$ using knowledge obtained in $\Pi$.

The problem studied here generalizes the one-dimensional version of transportability with limited scope and the multiple dimensional with unlimited scope previously studied. Interestingly, while certain effects might not be individually transportable to the target domain from the experiments in any of the available sources, combining different pieces from the various sources may enable their estimation. Conversely, it is also possible that effects are not estimable from multiple experiments in individual domains, but they are from experiments scattered throughout domains (discussed below).

The goal of this paper is to formally understand the conditions causal effects in the target domain are (non-parametrically) estimable from the available data. Sufficient conditions for "$mz$-transportability" were given in [13], but this treatment falls short of providing guarantees whether these conditions are also necessary, should be augmented, or even replaced by more general ones. This paper establishes the following results:

- A necessary and sufficient condition for deciding when causal effects in the target domain are estimable from both the statistical information available and the causal information transferred from the experiments in the domains.
- A proof that the algorithm proposed in [13] is in fact complete for computing the transport formula, that is, the strategy devised for combining the empirical evidence to synthesize the target relation cannot be improved upon.
- A proof that the do-calculus is complete for the $mz$-transportability class.

## 2 Background in Transportability

In this section, we consider other transportability instances and discuss the relationship with the $mz$-transportability setting. Consider Fig. 1(a) in which the node $S$ represents factors that produce differences between source and target populations. We conduct a randomized trial in Los Angeles (LA) and estimate the causal effect of treatment $X$ on outcome $Y$ for every age group $Z = z$, denoted by $P(y|do(x), z)$. We now wish to generalize the results to the population of New York City (NYC), but we find the distribution $P(x, y, z)$ in LA to be different from the one in NYC (call the latter $P^*(x, y, z)$). In particular, the average age in NYC is significantly higher than that in LA. How are we to estimate the causal effect of $X$ on $Y$ in NYC, denoted $R = P^*(y|do(x))$?[2][3]

The selection diagram – overlapping of the diagrams in LA and NYC – for this example (Fig. 1(a)) conveys the assumption that the *only* difference between the two populations are factors determining age distributions, shown as $S \to Z$, while age-specific effects $P^*(y|do(x), Z = z)$ are invariant across populations. Difference-generating factors are represented by a special set of variables called *selection variables* $S$ (or simply $S$-variables), which are graphically depicted as square nodes (■). From this assumption, the overall causal effect in NYC can be derived as follows:

$$
\begin{aligned}
R &= \sum_z P^*(y|do(x), z)P^*(z) \\
&= \sum_z P(y|do(x), z)P^*(z) \qquad (1)
\end{aligned}
$$

The last line constitutes a *transport formula* for $R$; it combines experimental results obtained in LA, $P(y|do(x), z)$, with observational aspects of NYC population, $P^*(z)$, to obtain a causal claim

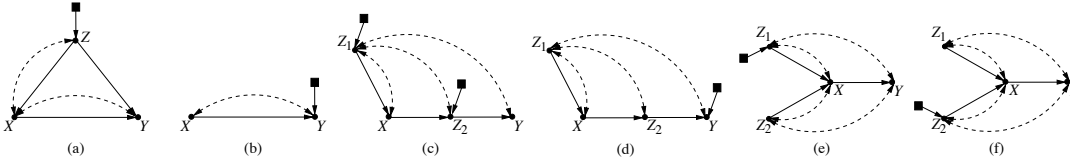

Figure 1: (a) Selection diagram illustrating when transportability of $R = P^*(y|do(x))$ between two domains is trivially solved through simple recalibration. (b) The smallest diagram in which a causal relation is not transportable. (c,d) Selection diagrams illustrating the impossibility of estimating $R$ through individual transportability from $\pi_a$ and $\pi_b$ even when $\mathbf{Z} = \{Z_1, Z_2\}$. If experiments over $\{Z_2\}$ is available in $\pi_a$ and over $\{Z_1\}$ in $\pi_b$, $R$ is transportable. (e,f) Selection diagrams illustrating opposite phenomenon – transportability through multiple domains is not feasible, but if $\mathbf{Z} = \{Z_1, Z_2\}$ in one domain is. The selection variables $S$ are depicted as square nodes (■).

$P^*(y|do(x))$ about NYC. In this trivial example, the transport formula amounts to a simple re-calibration (or re-weighting) of the age-specific effects to account for the new age distribution. In general, however, a more involved mixture of experimental and observational findings would be necessary to obtain an unbiased estimate of the target relation $R$. In certain cases there is no way to synthesize a transport formula, for instance, Fig. 1(b) depicts the smallest example in which transportability is not feasible (even with $X$ randomized). Our goal is to characterize these cases.

In real world applications, it may happen that only a limited amount of experimental information can be gathered at the source environment. The question arises whether an investigator in possession of a limited set of experiments would still be able to estimate the desired effects at the target domain. To illustrate some of the subtle issues that $mz$-transportability entails, consider Fig. 1(c,d) which concerns the transport of experimental results from two sources ($\{\pi_a, \pi_b\}$) to infer the effect of $X$ on $Y$ in $\pi^*$, $R = P^*(y|do(x))$. In these diagrams, $X$ may represent the treatment (e.g., cholesterol level), $Z_1$ represents a pre-treatment variable (e.g., diet), $Z_2$ represents an intermediate variable (e.g., biomarker), and $Y$ represents the outcome (e.g., heart failure). Assume that experimental studies randomizing $\{Z_2\}$ can be conducted in domain $\pi_a$ and $\{Z_1\}$ in domain $\pi_b$. A simple analysis can show that $R$ cannot be transported from either source alone (even when experiments are available over both variables) [9]. Still, combining experiments from both sources allows one to determine the effect in the target through the following transport formula [13]:

$$P^*(y|do(x)) \quad = \quad \sum_{z_2} P^{(b)}(z_2|x, do(Z_1)) P^{(a)}(y|do(z_2)) \tag{2}$$

This transport formula is a mixture of the experimental result over $\{Z_1\}$ from $\pi_b$, $P^{(b)}(z_2|x, do(Z_1))$, with the result of the experiment over $\{Z_2\}$ in $\pi_a$, $P^{(a)}(y|do(z_2))$, and constitute a consistent estimand of the target relation in $\pi^*$. Further consider Fig. 1(e,f) which illustrates the opposite phenomenon. In this case, if experiments over $\{Z_2\}$ are available in domain $\pi_a$ and over $\{Z_1\}$ in $\pi_b$, $R$ is not transportable. However, if $\{Z_1, Z_2\}$ are available in the same domain, say $\pi_a$, $R$ is transportable and equals $P^{(a)}(y|x, do(Z_1, Z_2))$, independently of the values of $Z_1$ and $Z_2$.

These intriguing results entail two fundamental issues that will be answered throughout this paper. First, whether the do-calculus is complete relative to such problems, that is, whether it would always find a transport formula whenever such exists. Second, assuming that there exists a sequence of applications of do-calculus that achieves the reduction required by $mz$-transportability, to find such a sequence may be computational intractable, so an efficient way is needed for obtaining such formula.

## 3  A Graphical Condition for $mz$-transportability

The basic semantical framework in our analysis rests on *structural causal models* as defined in [18, pp. 205], also called data-generating models. In the structural causal framework [18, Ch. 7], actions are modifications of functional relationships, and each action $do(\mathbf{x})$ on a causal model $M$ produces a new model $M_{\mathbf{x}} = \langle \mathbf{U}, \mathbf{V}, \mathbf{F_x}, P(\mathbf{U}) \rangle$, where $\mathbf{V}$ is the set of observable variables, $\mathbf{U}$ is the set of unobservable variables, and $\mathbf{F_x}$ is obtained after replacing $f_X \in \mathbf{F}$ for every $X \in \mathbf{X}$ with a new function that outputs a constant value $x$ given by $do(\mathbf{x})$.

We follow the conventions given in [18]. We denote variables by capital letters and their realized values by small letters. Similarly, sets of variables will be denoted by bold capital letters, sets

of realized values by bold small letters. We use the typical graph-theoretic terminology with the corresponding abbreviations $De(\mathbf{Y})_G$, $Pa(\mathbf{Y})_G$, and $An(\mathbf{Y})_G$, which will denote respectively the set of observable descendants, parents, and ancestors of the node set $\mathbf{Y}$ in $G$. A graph $G_{\mathbf{Y}}$ will denote the induced subgraph $G$ containing nodes in $\mathbf{Y}$ and all arrows between such nodes. Finally, $G_{\overline{\mathbf{X}}\underline{\mathbf{Z}}}$ stands for the edge subgraph of $G$ where all arrows incoming into $\mathbf{X}$ and all arrows outgoing from $\mathbf{Z}$ are removed.

Key to the analysis of transportability is the notion of *identifiability* [18, pp. 77], which expresses the requirement that causal effects are computable from a combination of non-experimental data $P$ and assumptions embodied in a causal diagram $G$. Causal models and their induced diagrams are associated with one particular domain (i.e., setting, population, environment), and this representation is extended in transportability to capture properties of two domains simultaneously. This is possible if we assume that the structural equations share the same set of arguments, though the functional forms of the equations may vary arbitrarily [7]. [4]

**Definition 1** (Selection Diagrams). *Let $\langle M, M^* \rangle$ be a pair of structural causal models relative to domains $\langle \pi, \pi^* \rangle$, sharing a diagram $G$. $\langle M, M^* \rangle$ is said to induce a selection diagram $D$ if $D$ is constructed as follows: every edge in $G$ is also an edge in $D$; $D$ contains an extra edge $S_i \rightarrow V_i$ whenever there might exist a discrepancy $f_i \neq f_i^*$ or $P(U_i) \neq P^*(U_i)$ between $M$ and $M^*$.*

In words, the $S$-variables locate the *mechanisms* where structural discrepancies between the two domains are suspected to take place.[5] Armed with the concept of identifiability and selection diagrams, $mz$-transportability of causal effects can be defined as follows [13]:

**Definition 2** ($mz$-Transportability). *Let $\mathcal{D} = \{D^{(1)}, ..., D^{(n)}\}$ be a collection of selection diagrams relative to source domains $\Pi = \{\pi_1, ..., \pi_n\}$, and target domain $\pi^*$, respectively, and $\mathbf{Z}_i$ (and $\mathbf{Z}^*$) be the variables in which experiments can be conducted in domain $\pi_i$ (and $\pi^*$). Let $\langle P^i, I_z^i \rangle$ be the pair of observational and interventional distributions of $\pi_i$, where $I_z^i = \bigcup_{\mathbf{Z}' \subseteq \mathbf{Z}_i} P^i(\mathbf{v}|do(\mathbf{z}'))$, and in an analogous manner, $\langle P^*, I_z^* \rangle$ be the observational and interventional distributions of $\pi^*$. The causal effect $R = P_{\mathbf{x}}^*(\mathbf{y})$ is said to be $mz$-transportable from $\Pi$ to $\pi^*$ in $\mathcal{D}$ if $P_{\mathbf{x}}^*(\mathbf{y})$ is uniquely computable from $\bigcup_{i=1,...,n} \langle P^i, I_z^i \rangle \cup \langle P^*, I_z^* \rangle$ in any model that induces $\mathcal{D}$.*

While this definition might appear convoluted, it is nothing more than a formalization of the statement "R need to be uniquely computable from the information set IS alone." Naturally, when IS has many components (multiple observational and interventional distributions), it becomes lengthy. This requirement of computability from $\langle P^*, I_z^* \rangle$ and $\langle P^i, I_z^i \rangle$ from all sources has a syntactic image in the do-calculus, which is captured by the following sufficient condition:

**Theorem 1** ([13]). *Let $\mathcal{D} = \{D^{(1)}, ..., D^{(n)}\}$ be a collection of selection diagrams relative to source domains $\Pi = \{\pi_1, ..., \pi_n\}$, and target domain $\pi^*$, respectively, and $\mathbf{S_i}$ represents the collection of $S$-variables in the selection diagram $D^{(i)}$. Let $\{\langle P^i, I_z^i \rangle\}$ and $\langle P^*, I_z^* \rangle$ be respectively the pairs of observational and interventional distributions in the sources $\Pi$ and target $\pi^*$. The effect $R = P^*(\mathbf{y}|do(\mathbf{x}))$ is $mz$-transportable from $\Pi$ to $\pi^*$ in $\mathcal{D}$ if the expression $P(\mathbf{y}|do(\mathbf{x}), \mathbf{S_1}, ..., \mathbf{S_n})$ is reducible, using the rules of the do-calculus, to an expression in which (1) do-operators that apply to subsets of $I_z^i$ have no $\mathbf{S_i}$-variables or (2) do-operators apply only to subsets of $I_z^*$.*

It is not difficult to see that in Fig. 1(c,d) (and also in Fig. 1(e,f)) a sequence of applications of the rules of do-calculus indeed reaches the reduction required by the theorem and yields a transport formula as shown in Section 2. It is not obvious, however, whether such sequence exists in Fig. 2(a,b) when experiments over $\{X\}$ are available in $\pi_a$ and $\{Z\}$ in $\pi_b$, and if it does not exist, it is also not clear whether this would imply the inability to transport. It turns out that in this specific example there is not such sequence and the target relation $R$ is not transportable, which means that there exist two models that are equally compatible with the data (i.e., both could generate the same dataset) while each model entails a different answer for the effect $R$ (violating the uniqueness requirement of Def. 2). [6] To demonstrate this fact formally, we show the existence of two structural

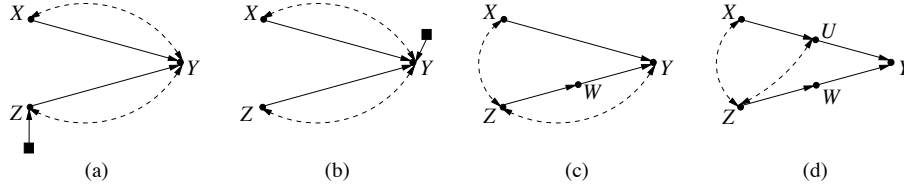

(a)        (b)        (c)        (d)

Figure 2: (a,b) Selection diagrams in which is not possible to transport $R = P^*(y|do(x))$ with experiments over $\{X\}$ in $\pi_a$ and $\{Z\}$ in $\pi_b$. (c,d) Example of diagrams in which some paths need to be extended for satisfying the definition of $mz^*$-shedge.

models $M_1$ and $M_2$ such that the following equalities and inequality between distributions hold,

$$
\begin{cases}
P_{M_1}^{(a)}(X,Z,Y) = P_{M_2}^{(a)}(X,Z,Y), \\
P_{M_1}^{(b)}(X,Z,Y) = P_{M_2}^{(b)}(X,Z,Y), \\
P_{M_1}^{(a)}(Z,Y|do(X)) = P_{M_2}^{(a)}(Z,Y|do(X)), \\
P_{M_1}^{(b)}(X,Y|do(Z)) = P_{M_2}^{(b)}(X,Y|do(Z)), \\
P_{M_1}^{*}(X,Z,Y) = P_{M_2}^{*}(X,Z,Y),
\end{cases}
\tag{3}
$$

for all values of $X$, $Z$, and $Y$, and

$$
P_{M_1}^{*}(Y|do(X)) \neq P_{M_2}^{*}(Y|do(X)),
\tag{4}
$$

for some value of $X$ and $Y$.

Let us assume that all variables in $\mathbf{U} \cup \mathbf{V}$ are binary. Let $U_1, U_2 \in \mathbf{U}$ be the common causes of $X$ and $Y$ and $Z$ and $Y$, respectively; let $U_3, U_4 \in \mathbf{U}$ be the random disturbances exclusive to $Z$ and $Y$, respectively, and $U_5, U_6 \in \mathbf{U}$ be extra random disturbances exclusive to $Y$. Let $S_a$ and $S_b$ index the model in the following way: the tuples $\langle S_a = 1, S_b = 0 \rangle$, $\langle S_a = 0, S_b = 1 \rangle$, $\langle S_a = 0, S_b = 0 \rangle$ represent domains $\pi_a$, $\pi_b$, and $\pi^*$, respectively. Define the two models as follows:

$$
M_1 = \begin{cases}
X = U_1 \\
Z = U_2 \oplus (U_3 \wedge S_a) \\
Y = ((X \oplus Z \oplus U_1 \oplus U_2 \oplus (U_4 \wedge S_b)) \\
\quad\quad \wedge U_5) + (\neg U_5 \wedge U_6)
\end{cases}
\quad
M_2 = \begin{cases}
X = U_1 \\
Z = U_2 \oplus (U_3 \wedge S_a) \\
Y = ((Z \oplus U_2 \oplus (U_4 \wedge S_b)) \\
\quad\quad \wedge U_5) \oplus (\neg U_5 \wedge U_6)
\end{cases}
$$

where $\oplus$ represents the *exclusive or* function. Both models agree in respect to $P(\mathbf{U})$, which is defined as $P(U_i) = 1/2$, $i = 1, ..., 6$. It is not difficult to evaluate these models and note that the constraints given in Eqs. (3) and (4) are indeed satisfied (including positivity), the result follows. [7]

Given that our goal is to demonstrate the converse of Theorem 1, we collect different examples of non-transportability, as the previous one, and try to make sense whether there is a pattern in such cases and how to generalize them towards a complete characterization of $mz$-transportability.

One syntactic subtask of $mz$-transportability is to determine whether certain effects are identifiable in some source domains where interventional data is available. There are two fundamental results developed for identifiability that will be relevant for $mz$-transportability as well. First, we should consider confounded components (or *c-components*), which were defined in [20] and stand for a cluster of variables connected through bidirected edges (which are not separable through the observables in the system). One key result is that each causal graph (and subgraphs) induces an unique C-component decomposition ([20, Lemma 11]). This decomposition was indeed instrumental for a series of conditions for ordinary identification [21] and the inability to recursively decompose a certain graph was later used to prove completeness.

**Definition 3** (C-component). *Let G be a causal diagram such that a subset of its bidirected arcs forms a spanning tree over all vertices in G. Then G is a C-component (confounded component).*

Subsequently, [22] proposed an extension of *C-components* called *C-forests*, essentially enforcing that each C-component has to be a spanning forest and closed under ancestral relations [20].

**Definition 4** (C-forest). *Let $G$ be a causal diagram where $\mathbf{Y}$ is the maximal root set. Then $G$ is a $\mathbf{Y}$-rooted C-forest if $G$ is a C-component and all observable nodes have at most one child.*

For concreteness, consider Fig. 1(c) and note that there exists a $C$-forest over nodes $\{Z_1, X, Z_2\}$ and rooted in $\{Z_2\}$. There exists another $C$-forest over nodes $\{Z_1, X, Z_2, Y\}$ rooted in $\{Y\}$. It is also the case that $\{Z_2\}$ and $\{Y\}$ are themselves trivial C-forests. When we have a pair of $C$-forests as $\{Z_1, X, Z_2\}$ and $\{Z_2\}$ or $\{Z_1, X, Z_2, Y\}$ and $\{Y\}$ – i.e., the root set does not intersect the treatment variables; these structures are called *hedges* and identifiability was shown to be infeasible whenever a hedge exists [22]. Clearly, despite the existence of hedges in Fig. 1(c,d), the effects of interest were shown to be $mz$-transportable. This example is an indication that hedges do not capture in an immediate way the structure needed for characterizing $mz$-transportability – i.e., a graph might be a hedge (or have a hedge as an edge sub–graph) but the target quantity might still be $mz$-transportable.

Based on these observations, we propose the following definition that may lead to the boundaries of the class of $mz$-transportable relations:

**Definition 5** ($mz^*$-shedge). *Let $\mathcal{D} = (D^{(1)}, \ldots, D^{(n)})$ be a collection of selection diagrams relative to source domains $\Pi = (\pi_1, \ldots, \pi_n)$ and target domain $\pi^*$, respectively, $\mathbf{S_i}$ represents the collection of S-variables in the selection diagram $D^{(i)}$, and let $D^{(*)}$ be the causal diagram of $\pi^*$. Let $\{\langle P^i, I_z^i \rangle\}$ be the collection of pairs of observational and interventional distributions of $\{\pi_i\}$, where $I_z^i = \bigcup_{\mathbf{Z}' \subseteq \mathbf{Z}_i} P^i(\mathbf{v}|do(\mathbf{z}'))$, and in an analogous manner, $\langle P^*, I_z^* \rangle$ be the observational and interventional distributions of $\pi^*$, for $\mathbf{Z_i}$ the set of experimental variables in $\pi_i$. Consider a pair of $\mathbf{R}$-rooted C-forests $\mathcal{F} = \langle F, F' \rangle$ such that $F' \subset F$, $F' \cap \mathbf{X} = \emptyset$, $F \cap \mathbf{X} \neq \emptyset$, and $\mathbf{R} \subseteq An(\mathbf{Y})_{G_{\overline{\mathbf{X}}}}$ (called a hedge [22]). We say that the induced collection of pairs of $\mathbf{R}$-rooted C-forests over each diagram, $\langle \mathcal{F}^{(*)}, \mathcal{F}^{(1)}, ..., \mathcal{F}^{(n)} \rangle$, is an $mz$-shedge for $P_{\mathbf{x}}^*(\mathbf{y})$ relative to experiments $(I_z^*, I_z^1, ..., I_z^n)$ if they are all hedges and one of the following conditions hold for each domain $\pi_i$, $i = \{*, 1, ..., n\}$:*

1. *There exists at least one variable of $\mathbf{S_i}$ pointing to the induced diagram $F'^{(i)}$, or*

2. *$(F^{(i)} \setminus F'^{(i)}) \cap \mathbf{Z_i}$ is an empty set, or*

3. *The collection of pairs of C-forests induced over diagrams, $\langle \mathcal{F}^{(*)}, \mathcal{F}^{(1)}, \ldots, F^{(i)} \setminus \mathbf{Z_i^*}, \ldots, \mathcal{F}^{(n)} \rangle$, is also an $mz$-shedge relative to $(I_z^*, I_z^1, ..., I_{z \setminus z_i^*}^i, ..., I_z^n)$, where $\mathbf{Z_i^*} = (F^{(i)} \setminus F'^{(i)}) \cap \mathbf{Z_i}$.*

*Furthermore, we call $mz^*$-shedge the $mz$-shedge in which there exist one directed path from $\mathbf{R} \setminus (\mathbf{R} \cap De(\mathbf{X})_F)$ to $(\mathbf{R} \cap De(\mathbf{X})_F)$ not passing through $\mathbf{X}$ (see also appendix 3).*

The definition of $mz^*$-shedge might appear involved, but it is nothing more than the articulation of the computability requirement of Def. 2 (and implicitly the syntactic goal of Thm. 1) in a more explicit graphical fashion. Specifically, for a certain factor $Q_i^*$ needed for the computation of the effect $Q^* = P^*(\mathbf{y}|do(\mathbf{x}))$, in at least one domain, (i) it should be enforced that the $S$-nodes are separable from the inducing root set of the component in which $Q_i^*$ belongs, and further, (ii) the experiments available in this domain are sufficient for solving $Q_i^*$. For instance, assuming we want to compute $Q^* = P^*(y|do(x))$ in Fig. 1(c, d), $Q^*$ can be decomposed into two factors, $Q_1^* = P_{z_1,x}^*(z_2)$ and $Q_2^* = P_{z_1,x,z_2}^*(y)$. It is the case that for factor $Q_1^*$, (i) holds true in $\pi_b$ and (ii) the experiments available over $Z_1$ are enough to guarantee the computability of this factor (similar analysis applies to $Q_2^*$) – i.e., there is no $mz^*$-shedge and $Q^*$ is computable from the available data.

Def. 5 also asks for the explicit existence of a path from the nodes in the root set $\mathbf{R} \setminus (\mathbf{R} \cap De(\mathbf{X})_F)$ to $(\mathbf{R} \cap De(\mathbf{X})_F)$, a simple example can help to illustrate this requirement. Consider Fig. 2(c) and the goal of computing $Q = P^*(y|do(x))$ without extra experimental information. There exists a hedge for $Q$ induced over $\{X, Z, Y\}$ without the node $W$ (note that $\{W\}$ is a c-component itself) and the induced graph $G_{\{X,Z,Y\}}$ indeed leads to a counter-example for the computability of $P^*(z, y|do(x))$. Using this subgraph alone, however, it would not be possible to construct a counter-example for the marginal effect $P^*(y|do(x))$. Despite the fact that $P^*(z, y|do(x))$ is not computable from $P^*(x, z, y)$, the quantity $P^*(y|do(x))$ is identifiable in $G_{\{X,Z,Y\}}$, and so any structural model compatible with this subgraph will generate the same value under the marginalization over $Z$ from $P^*(z, y|do(x))$. Also, it might happen that the root set $\mathbf{R}$ must be augmented (Fig. 2(d)), so we prefer to add this requirement explicitly to the definition. (There are more involved scenarios that

PROCEDURE $\mathbf{TR^{mz}}(\mathbf{y}, \mathbf{x}, \mathcal{P}, \mathcal{I}, \mathcal{S}, \mathcal{W}, D)$

INPUT: $\mathbf{x}, \mathbf{y}$: value assignments; $\mathcal{P}$: local distribution relative to domain $\mathcal{S}$ ($\mathcal{S} = 0$ indexes $\pi^*$) and active experiments $\mathcal{I}$; $\mathcal{W}$: weighting scheme; $D$: backbone of selection diagram; $\mathbf{S_i}$: selection nodes in $\pi_i$ ($\mathbf{S_0} = \emptyset$ relative to $\pi^*$); [The following set and distributions are globally defined: $\mathbf{Z}_i, P^*, P_{\mathbf{Z}_i}^{(i)}$.]

OUTPUT: $P_{\mathbf{x}}^*(\mathbf{y})$ in terms of $P^*, P_{\mathbf{Z}}^*, P_{\mathbf{Z}_i}^{(i)}$ or $FAIL(D, C_0)$.

1   **if** $\mathbf{x} = \emptyset$, **return** $\sum_{\mathbf{V} \setminus \mathbf{Y}} \mathcal{P}$.
2   **if** $V \setminus An(\mathbf{Y})_D \neq \emptyset$, **return** $\mathbf{TR^{mz}}(\mathbf{y}, \mathbf{x} \cap An(\mathbf{Y})_D, \sum_{\mathbf{V} \setminus An(\mathbf{Y})_D} \mathcal{P}, \mathcal{I}, \mathcal{S}, \mathcal{W}, D_{An(\mathbf{Y})})$.
3   set $\mathbf{W} = (\mathbf{V} \setminus \mathbf{X}) \setminus An(\mathbf{Y})_{D_{\overline{\mathbf{X}}}}$.
    **if** $\mathbf{W} \neq \emptyset$, **return** $\mathbf{TR^{mz}}(\mathbf{y}, \mathbf{x} \cup \mathbf{w}, \mathcal{P}, \mathcal{I}, \mathcal{S}, \mathcal{W}, D)$.
4   **if** $\mathcal{C}(D \setminus \mathbf{X}) = \{C_0, C_1, ..., C_k\}$, **return** $\sum_{\mathbf{V} \setminus \{\mathbf{Y}, \mathbf{X}\}} \prod_i \mathbf{TR^{mz}}(c_i, \mathbf{v} \setminus c_i, \mathcal{P}, \mathcal{I}, \mathcal{S}, \mathcal{W}, D)$.
5   **if** $\mathcal{C}(D \setminus \mathbf{X}) = \{C_0\}$,
6       **if** $\mathcal{C}(D) \neq \{D\}$,
7           **if** $C_0 \in \mathcal{C}(D)$, **return** $\prod_{i | V_i \in C_0} \sum_{\mathbf{V} \setminus V_D^{(i)}} \mathcal{P} / \sum_{\mathbf{V} \setminus V_D^{(i-1)}} \mathcal{P}$.
8           **if** $(\exists C')C_0 \subset C' \in \mathcal{C}(D)$,
                **for** $\{i | V_i \in C'\}$, **set** $\kappa_{\mathbf{i}} = \kappa_i \cup v_D^{(i-1)} \setminus C'$.
                **return** $\mathbf{TR^{mz}}(\mathbf{y}, \mathbf{x} \cap C', \prod_{i | V_i \in C'} \mathcal{P}(V_i | V_D^{(i-1)} \cap C', \kappa_{\mathbf{i}}), \mathcal{I}, \mathcal{S}, \mathcal{W}, C')$.
9       **else**,
10          **if** $\mathcal{I} = \emptyset$, **for** $i = 0, ..., |\mathcal{D}|$,
                **if** $((\mathbf{S_i} \perp\!\!\!\perp \mathbf{Y} \mid \mathbf{X})_{D_{\overline{\mathbf{X}}}^{(i)}} \wedge (\mathbf{Z_i} \cap \mathbf{X} \neq \emptyset))$, $E_i = \mathbf{TR^{mz}}(\mathbf{y}, \mathbf{x} \setminus \mathbf{z_i}, \mathcal{P}, \mathbf{Z_i} \cap \mathbf{X}, i, \mathcal{W}, D \setminus \{\mathbf{Z_i} \cap \mathbf{X}\})$.
11          **if** $|\mathbf{E}| > 0$, **return** $\sum_{i=1}^{|\mathbf{E}|} w_i^{(j)} E_i$.
12          **else**, **FAIL**$(D, C_0)$.

Figure 3: Modified version of identification algorithm capable of recognizing $mz$-transportability.

we prefer to omit for the sake of presentation.) After adding the directed path from $Z$ to $Y$ that passes through $W$, we can construct the following counter-example for $Q$:

$$M_1 = \begin{cases} X = U_1 \\ Z = U_1 \oplus U_2 \\ W = ((Z \oplus U_3) \vee B) \oplus (B \wedge (1 \oplus Z)) \\ Y = ((X \oplus W \oplus U_2) \wedge A) \\ \quad \oplus (A \vee (1 \oplus X \oplus W \oplus U2)), \end{cases} \quad M_2 = \begin{cases} X = U_1 \\ Z = U_2 \\ W = ((Z \oplus U_3) \vee B) \oplus (B \wedge (1 \oplus Z)) \\ Y = ((W \oplus U_2) \wedge A) \\ \quad \oplus (A \vee (1 \oplus W \oplus U2)), \end{cases}$$

with $P(U_i) = 1/2, \forall i, P(A) = P(B) = 1/2$. It is not immediate to show that the two models produce the desired property. Refer to Appendix 2 for a formal proof of this statement.

Given that the definition of $mz^*$-shedge is justified and well-understood, we can now state the connection between hedges and $mz^*$-shedges more directly (the proof can be found in Appendix 3):

**Theorem 2.** *If there is a hedge for $P_{\mathbf{x}}^*(\mathbf{y})$ in $G$ and no experimental data is available (i.e., $I_z^* = \{\}$), there exists an $mz^*$-shedge for $P_{\mathbf{x}}^*(\mathbf{y})$ in $G$.*

Whenever one domain is considered and no experimental data is available, this result states that a $mz^*$-shedge can always be constructed from a hedge, which implies that we can operate with $mz^*$-shedges from now on (the converse holds for $Z = \{\}$). Finally, we can concentrate on the most general case of $mz^*$-shedges with experimental data in multiple domains as stated in the sequel:

**Theorem 3.** *Let $\mathcal{D} = \{D^{(1)}, ..., D^{(n)}\}$ be a collection of selection diagrams relative to source domains $\Pi = \{\pi_1, ..., \pi_n\}$, and target domain $\pi^*$, respectively, and $\{I_z^i\}$, for $i = \{*, 1, ..., n\}$ defined appropriately. If there is an $mz^*$-shedge for the effect $R = P_{\mathbf{x}}^*(\mathbf{y})$ relative to experiments $(I_z^*, I_z^1, ..., I_z^n)$ in $\mathcal{D}$, $R$ is not $mz$-transportable from $\Pi$ to $\pi^*$ in $\mathcal{D}$.*

This is a powerful result that states that the existence of a $mz^*$-shedge precludes $mz$-transportability. (The proof of this statement is somewhat involved, see the supplementary material for more details.) For concreteness, let us consider the selection diagrams $\mathcal{D} = (D^{(a)}, D^{(b)})$ relative to domains $\pi_a$ and $\pi_b$ in Fig. 2(a,b). Our goal is to $mz$-transport $Q = P^*(y|do(x))$ with experiments over $\{X\}$ in $\pi_a$ and $\{Z\}$ in $\pi_b$. It is the case that there exists an $mz^*$-shedge relative to the given experiments. To witness, first note that $F' = \{Y, Z\}$ and $F = F' \cup \{X\}$, and also that there exists a selection variable $S$ pointing to $F'$ in both domains – the first condition of Def. 5 is satisfied. This is a trivial graph with 3 variables that can be solved by inspection, but it is somewhat involved to efficiently evaluate the conditions of the definition in more intricate structures, which motivates the search for a procedure for recognizing $mz^*$-shedges that can be coupled with the previous theorem.

# 4 Complete Algorithm for $mz$-transportability

There exists an extensive literature concerned with the problem of computability of causal relations from a combination of assumptions and data [21, 22, 7, 13]. In this section, we build on the works that treat this problem by graphical means, and we concentrate particularly in the algorithm called $\mathbf{TR^{mz}}$ constructed in [13] (see Fig. 3) that followed some of the results in [21, 22, 7].

The algorithm $\mathbf{TR^{mz}}$ takes as input a collection of selection diagrams with the corresponding experimental data from the corresponding domains, and it returns a transport formula whenever it is able to produce one. The *main idea* of the algorithm is to leverage the c-component factorization [20] and recursively decompose the target relation into manageable pieces (line 4), so as to try to solve each of them separately. Whenever this standard evaluation fails in the target domain $\pi^*$ (line 6), $\mathbf{TR^{mz}}$ tries to use the experimental information available from the target and source domains (line 10). (For a concrete view of how $\mathbf{TR^{mz}}$ works, see the running example in [13, pp. 7]. )

In a systematic fashion, the algorithm basically implements the declarative condition delineated in Theorem 1. $\mathbf{TR^{mz}}$ was shown to be sound [13, Thm. 3], but there is no theoretical guarantee on whether failure in finding a transport formula implies its non-existence and perhaps, the complete lack of transportability. This guarantee is precisely what we state in the sequel.

**Theorem 4.** *Assume $\mathbf{TR^{mz}}$ fails to transport the effect $P^*_{\mathbf{x}}(\mathbf{y})$ (exits with failure executing line 12). Then there exists $\mathbf{X}' \subseteq \mathbf{X}$, $\mathbf{Y}' \subseteq \mathbf{Y}$, such that the graph pair $D, C_0$ returned by the fail condition of $\mathbf{TR^{mz}}$ contains as edge subgraphs C-forests F, F' that span a $mz^*$-shedge for $P^*_{\mathbf{x}'}(\mathbf{y}')$.*

*Proof.* Let $D$ be the subgraph local to the call in which $\mathbf{TR^{mz}}$ failed, and $\mathbf{R}$ be the root set of $D$. It is possible to remove some directed arrows from $D$ while preserving $\mathbf{R}$ as root, which result in a $\mathbf{R}$-rooted c-forest $F$. Since by construction $F' = F \cap C_0$ is closed under descendents and only directed arrows were removed, both $F, F'$ are C-forests. Also by construction $\mathbf{R} \subset An(\mathbf{Y})_{G_{\overline{\mathbf{X}}}}$ together with the fact that $\mathbf{X}$ and $\mathbf{Y}$ from the recursive call are clearly subsets of the original input. Before failure, $\mathbf{TR^{mz}}$ evaluated false consecutively at lines 6, 10, and 11, and it is not difficult to see that an $S$-node points to $F'$ or the respective experiments were not able to break the local hedge (lines 10 and 11). It remains to be showed that this $mz$-shedge can be stretched to generate a $mz^*$-shedge, but now the same construction given in Thm. 2 can be applied (see also supplementary material). □

Finally, we are ready to state the completeness of the algorithm and the graphical condition.

**Theorem 5** (completeness). $\mathbf{TR^{mz}}$ *is complete.*

**Corollary 1** ($mz^*$-shedge characterization). $P^*_{\mathbf{x}}(\mathbf{y})$ *is $mz$-transportable from $\Pi$ to $\pi^*$ in $\mathcal{D}$ if and only if there is not $mz^*$-shedge for $P_{\mathbf{x}'}(\mathbf{y}')$ in $\mathcal{D}$ for any $\mathbf{X}' \subseteq \mathbf{X}$ and $\mathbf{Y}' \subseteq \mathbf{Y}$.*

Furthermore, we show below that the do-calculus is complete for establishing $mz$-transportability, which means that failure in the exhaustive application of its rules implies the non-existence of a mapping from the available data to the target relation (i.e., there is no $mz$-transport formula), independently of the method used to obtain such mapping.

**Corollary 2** (do-calculus characterization). *The rules of do-calculus together with standard probability manipulations are complete for establishing $mz$-transportability of causal effects.*

# 5 Conclusions

In this paper, we provided a complete characterization in the form of a graphical condition for deciding $mz$-transportability. We further showed that the procedure introduced in [1] for computing the transport formula is complete, which means that the set of transportable instances identified by the algorithm cannot be broadened without strengthening the assumptions. Finally, we showed that the do-calculus is complete for this class of problems, which means that finding a proof strategy in this language suffices to solve the problem. The non-parametric characterization established in this paper gives rise to a new set of research questions. While our analysis aimed at achieving unbiased transport under asymptotic conditions, additional considerations need to be taken into account when dealing with finite samples. Specifically, when sample sizes vary significantly across studies, statistical power considerations need to be invoked along with bias considerations. Furthermore, when no transport formula exists, approximation techniques must be resorted to, for example, replacing the requirement of non-parametric analysis with assumptions about linearity or monotonicity of certain relationships in the domains. The nonparametric characterization provided in this paper should serve as a guideline for such approximation schemes.

## Footnotes

[1]Traditionally, the machine learning literature has been concerned about discrepancies among domains in the context, almost exclusively, of predictive or classification tasks as opposed to learning causal or counterfactual measures [14, 15]. Interestingly, recent work on anticausal learning leverages knowledge about invariances of the underlying data-generating structure across domains, moving the literature towards more general modalities of learning [16, 17].

[2]We will use $P_{\mathbf{x}}(\mathbf{y} \mid \mathbf{z})$ interchangeably with $P(\mathbf{y} \mid do(\mathbf{x}), \mathbf{z})$.

[3]We use the structural interpretation of causal diagrams as described in [18, pp. 205] (see also Appendix 1).

[4]As discussed in the reference, the assumption of no structural changes between domains can be relaxed, but some structural assumptions regarding the discrepancies between domains must still hold (e.g., acyclicity).

[5]Transportability assumes that enough structural knowledge about both domains is known in order to substantiate the production of their respective causal diagrams. In the absence of such knowledge, *causal discovery* algorithms might be used to infer the diagrams from data [19, 18].

[6]This is usually an indication that the current state of scientific knowledge about the problem (encoded in the form of a selection diagram) does not constrain the observed distributions in such a way that an answer is entailed independently of the details of the functions and probability over the exogenous.

[7]To a more sophisticated argument on how to evaluate these models, see proofs in appendix 3.

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
