[Reviews · NeurIPS 2014]

Submitted by Assigned_Reviewer_11

The authors address the problem (called mz-transportability) of inferring a causal relationship R in a target domain using causal knowledge obtained in multiple source domains with limited experiments.
In the target domain only passive observations are available (or also some limited causal knowledge) which is not enough alone for the inference of R.

Specifically the contributions are:
1. A necessary and sufficient condition for deciding when causal effects in the target domain are estimable from the available information.
2. A proof that a previously proposed algorithm (TR^mz) for computing the transport formula is complete.
3. A proof that the do-calculus is complete for establishing mz-transportability of causal effects.

*Quality & Clarity:
This is mainly a theoretical paper and the claims are well-supported by the authors.
It is well-organized and well-presented. However, the claims and proofs are often quite involved (as also pointed out by the authors) and I found it quite difficult to follow some parts (for example page 6) and fully understand the details. Nevertheless, I would probably attribute this more to the advanced theoretical analysis rather than to the lack of good presentation of the paper.

I would propose for improvement:
- Define what is "a maximal root set" in Definition 4 (like in [21]).
- In Def.5: "(called an hedge)" -> "(called an hedge for P_x(y))".
- The definition of an hedge after Definition 4 is confusing. You could include a more rigorous definition of what is an hedge there, instead of including it in Definition 5.
- What do you mean by "separable"?
- It would be beneficial for the reader to present some parts in a simpler way, if possible.

*Originality & Significance:
The paper builds on the NIPS 2013 paper ("[1] Transportability from Multiple Environments with Limited Experiments") which is adequately referenced. [1] included necessary OR sufficient conditions for transportability and proved only soundness of TR^mz. The current paper advances over [1] by providing conditions for complete characterization of transportability, proving that TR^mz is complete and proving that do-calculus is complete for the mz-transportability class. So, I consider the paper having a significant contribution.

To be more explicit, you could add in the beginning of section 3 that the first part of this section (page 4) repeats from [1] or other publications (not only referring to them in page 4). Also, for eq. (2), add a reference to [1], where it is explain how eq. (2) is acquired.

Some more comments:
- Abstract, 3rd line: it seems that "limited experimental data" only refers to the target domain and not to the sources. Consider changing appropriately.
- Motivation, 1st line: maybe missing a "that" after "experiments"? Consider rewriting sentence.
- p.2, par.4: "the goal of....conditions the causal effect" -> "...conditions causal effects in the target domain"
- First bullet before section 2: last line: "in the domains" -> "in the source domains"?
- Section 2, line 2: you refer to the node S of Fig.1(a) which is missing: Consider adding the letter S in this square node of the figure.
- Footnote 1: "..exclusively, on.." -> "..exclusively, of.."
- Footnote 1: along with refs [13,14] you could also consider some of the following references:
Ben-David et al. Analysis of representations for domain adaptation
Mansour et al. Domain adaptation: learning bounds and algorithms
Mansour et al. Domain adaptation with multiple sources
Ben-David et al. A theory of learning from different domains
- p.3, last line of 1st par: period before "our goal is"
- explain what is V, U etc. in M_x
- what is "IS" after Definition 2?
- After Theorem 1: "It is not difficult to see..": add reference to [1]
- middle p.5: period before "the result follows"
- first line after Definition 4: remove "and" before "rooted"
- p.7 middle: period before "refer to Appendix 2.."
- Theorem 4:, last line: "contain" ->"contains", "spans"->"span", "a" ->"an" and "F'". What are sC-forests?
- proof of Thm 4: don't you need capital C for c-forest?,
"shows" -> "showed"
- After Thm 4 write that the proof is provided in the supp. material.
- Maybe it would be better to use pi_c instead of pi_a for the source of Fig.1(c).
- Supplementary, last page:
-- proof of Thm 5: remove one "the" (first line), remove one full stop (least line)
-- Proof of Corollary 1: "...from the previous Corollary": do you mean from the previous Theorem?
Summary: I consider the contribution of this paper significant for NIPS, even though the definitions, theorems and proofs are often quite involved, making it difficult to understand all details.

Submitted by Assigned_Reviewer_16

This paper is concerned with the problem of transportability, which is a generalization of the problem of identifiability of causal effects given causal assumptions or knowledge encoded in a causal diagram. A paper published in NIPS last year by Bareinboim et al. tackled a quite general version of the problem. Suppose we observe a set of variables in a number of domains, a target domain and one or more source domains, all of which are assumed to share a (known) qualitative causal structure over the observed variables and some latent variables. The structural equations possibly differ between a source domain and the target domain, as indicated by selection variables in the so-called selection diagrams, which are also assumed to be known. Moreover, suppose in each domain, a subset of the observed variables can be actively controlled to collect interventional data (limited experiments). The question is whether or not a certain interventional probability in the target domain is (non-parametrically) identifiable from the available observational and interventional distributions in all the domains. The paper under review presents further results on the problem, showing that the sound algorithm presented in Bareinboim et al.’s paper in last year’s NIPS is actually complete and that Pearl’s do-calculus is also complete with regard to this problem.

The new results are useful contributions to a well-defined and interesting line of research, which has to do with such important topics as “external validity” and “meta-analysis”. Part of the paper is quite dense and probably not very readable for readers who are not familiar with previous work, but the main ideas are sufficiently clear. I have a few comments/questions:

1. In Definition 1, Pearl’s definition of structural equation models is implicitly taken in which each structural equation is associated with one and only one U-variable, i.e., that the structural equation f_i (for variable V_i) is associated with U_i. It is thus a little confusing when several U-variables appear in the subsequent examples on p. 5 and elsewhere.

2. An assumption made in Definition 2 is that if a set Z can be subject to an intervention in a domain, then any subset of Z can be subject to an intervention (without affecting other variables in Z) in that domain. Can this assumption be relaxed?

3. I understand that Theorem 1 is from Bareinboim et al. (2013), but I found the statement a little odd. What does it mean to say that "a do-operator has no S_i-variables"? Does it mean that the do-operator does not apply to any S_i-variable? But if a do-operator applies to a subset of I_z^i, doesn’t it automatically follow that it does not apply to any S_i-variable? Also, it seems an expression in which every do-operator applies to some set other than subsets of I_z^i would satisfy (1).

4. In footnote 5, it is suggested that causal discovery methods might be used to learn the causal diagrams. It may be worth stressing that the causal discovery algorithms can rarely, if ever, learn the full causal diagram with latent variables. Also, I am curious how selection diagrams are to be learned from data.

5. Definition 5 is the key in the paper, but its statement and the subsequent remarks are quite dense. Perhaps it will be easier to read if the definition of “hedge” is first given separately, and a “mz-shedge” is then defined as a collection of hedges that satisfy certain conditions.

In the subsequent paragraph, it will probably help if the conditions 1, 2, and 3 in the definition can be explicitly linked to conditions (i) and (ii).

The remark at the bottom of the page “Also, it might happen that the root set R must be augmented …” is elusive. I would appreciate a clarification of what it means.
Summary: The paper establishes new, useful results on an interesting problem.

Submitted by Assigned_Reviewer_39

This paper address the necessary and sufficient condition for deciding the feasibility of mz-transportability. mz-transportability is a property of a causal effect which relation in a target domain can be uniquely computed from its relations in different domains. This idea is considered to provide a basis of scientific discovery in unexplored domain based on the knowledge on our well-explored domains. The key consequence of this paper is the completeness of the do-calculus, i.e., there exists a set of the control operations of variables to judge any mz-transportability, and they showed that an algorithm presented in a past study to recognize mz-transportability is complete.

Quality
The theoretical analysis and its consequences presented in this paper seems valid. Though this is theoretical work, the proofs of many theorems are not provided in the main body of this paper. But, the lack of the proofs is acceptable, since these proofs are too lengthy to fit in the paper. Except this point, theoretical key points are well articulated.
A drawback is that the feasibility and the tractability of the actual implementation of the algorithm are not clear. Yet, the outcome on the completeness of the do-calculus across multiple domains is very important result, and will provide a basis of the future causal inference study.

Clarity
The paper is well written. Main claims of the authors are well described. As I pointed out above,the theoretical key pints are well articulated. The given set of the theorems and the corollaries seem appropriate.
But, the notations and the explanations in this paper seem too involved in the field of causal inference. The readers unfamiliar with the field will have many difficulties to understand them. In this regard, some descriptions in the paper should be even more well designed. I see some rooms to improve the explanations in the paper including the following points.
* 4th line of Section 3
Mx = (U,V,Fx,P(U)) suddenly appears without any explanation of the notation. I see some explanation on V on page 5; "Let V be the set of observable ....". But, including V, this formulation should be comprehensively described at the top of the section 3.
* In definition 1
"D contains an extra edge from Si to Vi". This is a description of an edge without explaining about Si. What is "S-variables"? It is unclear. To me, it seems unobserved exogenous variable or noise in the latter explanation. But, it should be more comprehensively explained around this definition.

Originality
The idea presented in this paper seems an extension of the work of J. Pearl et al.'s papers recently presented in AAAI, NIPS and AISTATS. In this regard, this work is rather incremental. However, the completeness of the do-calculus in terms of the mz-transportability across multiple domains provides a theoretical basis of causal inference. Its content to originally clarify this feature of the calculus is attractive. In addition, the proposal of mz*-shedge, which plays a central role to derive the completeness and the associated algorithms, is original.

Significance
The idea of mz*-shedge and the completeness of the do-calculus in terms of the mz-transportability have a potentially large contribution to the future study of causal inference.
Summary: The description of this paper is rather involved in the field of causal inference. But, the idea proposed and presented in this paper will provide a fundamental basis of the many future studies of causal inference.
Author Feedback
Author rebuttal: We thank the reviewers for their very valuable comments, suggestions, and questions, we plan to improve the final text incorporating these observations accordingly. We briefly address some of the issues below.

Reviewer 11: Maximality means that we cannot add variables or edges without destroying a certain property, in this specific context we cannot stretch the root set without violating the constraints imposed by the definition of c-components; we will add a note explaining this point.

Reviewer 16,Q2: We assume that all subsets of the variables can be intervened upon and the question of how to relax this requirement is certainly an interesting one.

Q3: This is certainly a relevant issue, we will clarify this in the paper.
a) It is assumed in the paper that we cannot intervene on S-variables since they are unobservables, which is essentially the same requirement imposed over the latent variables U assumed throughout the causal inference literature.

b) The “separability condition" is not 'automatically' satisfied as shown in Fig. 1(b).

c) This condition means that evaluating transportability of the sub-expression P*(Yk | do(Xl), Zm) in the target domain can be translated to judging independence statements relative to an equivalent expression P*(Yk | do(Xl), Zm, S1, S2, S3, …) (as defined in Bareinboim et al. (2013)) — more specifically, whether Yk is independent of Si given {Xl, Zm} in the respective mutilated graph for at least one set Si (selection diagram in domain \Pi_i).

Q5. We show in Fig. 2(d) an example in which the root set needs to be augmented. In this case, the original root set of the respective hedge is {Z, U}, but the construction needs to be extended to reach Y in order to characterize non-mz transportability (it is not possible to do so without stretching the root set). This is an involved matter, we will try to improve the manuscript regarding this issue or perhaps even move the whole discussion to the appendix.

Reviewer 39: Assumptions are encoded in a selection diagram through the removal of S-variables from the graph (analogous to probabilistic invariances encoded through the removal of arrows in an ordinary graphical model). More specifically, for a given observable variable Vi, the removal of the corresponding S-variable encodes that both the function and the probability of the exogenous variables affecting Vi are the same. In the extreme case, S-variables could be added to point to all observable variables in the system, which means that one does not want to make any assumption about invariances across domains (i.e., the two domains are unrelated). This is a very fundamental point and we will add a more explicit explanation in the manuscript.

We appreciate the reviewers' feedback and encouragement to pursue this line of research.